Litsea glaucescens Kuth possesses bactericidal activity against Listeria monocytogenes

Gress-Antonio Carlos David 1
Rivero-Perez Nallely 1
Marquina-Bahena Silvia 2
Alvarez Laura 2
Zaragoza-Bastida Adrian 1
Martínez-Juárez Víctor Manuel 1
Sosa-Gutierrez Carolina G. 1
Ocampo-López Juan 1
Zepeda-Bastida Armando 1
Ojeda-Ramírez Deyanira dojeda@uaeh.edu.mx 1
1 Área Académica de Medicina Veterinaria y Zootecnia, Universidad Autónoma del Estado de Hidalgo , Tulancingo de Bravo , Hidalgo , Mexico
2 Centro de Investigaciones Químicas, Universidad Autónoma del Estado de Morelos , Cuernavaca , Morelos , Mexico
Puia Zothan
Electronic publication date: 2023 Dec 1
Publication date: 2023
Volume: 11
Electronic Location ID: e16522
Received 2023 Jun 29; Accepted 2023 Nov 3
Copyright: ©2023 Gress-Antonio et al.
Copyright year: 2023
Copyright holder: Gress-Antonio et al.
License: This is an open access article distributed under the terms of the Creative Commons Attribution License, which permits unrestricted use, distribution, reproduction and adaptation in any medium and for any purpose provided that it is properly attributed. For attribution, the original author(s), title, publication source (PeerJ) and either DOI or URL of the article must be cited.
License URL: https://creativecommons.org/licenses/by/4.0/

Keywords: Litsea glaucescens, Listeria monocytogenes, Pinocembrin, Bactericidal activity

Funding: CONAHCYT 416289 The Laboratorio Nacional de Estructura de Macromoléculas (CONAHCYT 294406) supported the spectroscopic analysis Carlos David Gress-Antonio was awarded fellowship No. 416289 by CONAHCYT. The Laboratorio Nacional de Estructura de Macromoléculas (CONAHCYT 294406) supported the spectroscopic analysis. The funders had no role in study design, data collection and analysis, decision to publish, or preparation of the manuscript.

==============================
Background

Litsea glaucencens Kuth is an aromatic plant used for food seasoning food and in Mexican traditional medicine. Among, L. glaucencens leaves properties, it has proven antibacterial activity which can be used against opportunistic pathogens like Listeria monocytogenes, a foodborne bacteria that is the causal agent of listeriosis, a disease that can be fatal in susceptible individuals. The aim of this work was to investigate the antibacterial activity of L. glaucescens Kuth leaf extracts against L. monocytogenes and to identify its bioactive components.

Material and Methods

L. glaucences leaves were macerated with four solvents of different polarity (n-hexane, dichloromethane, ethyl acetate, and methanol). To determine the capacity to inhibit bacterial proliferation in vitro, agar diffusion and microdilution methods were used. Next, we determined the minimal bactericidal concentration (MBC). Finally, we determined the ratio of MBC/MIC. Metabolites present in the active methanolic extract from L. glaucescens Kuth (LgMeOH) were purified by normal-phase open column chromatography. The structure of the antibacterial metabolite was determined using nuclear magnetic resonance (1H, 13C, COSY, HSQC) and by comparison with known compounds.

Results

The LgMeOH extract was used to purify the compound responsible for the observed antimicrobial activity. This compound was identified as 5,7-dihydroxyflavanone (pinocembrin) by analysis of its spectroscopic data and comparison with those described. The MIC and MBC values obtained for pinocembrin were 0.68 mg/mL, and the ratio MBC/MIC for both LgMeOH and pinocembrin was one, which indicates bactericidal activity.

Conclusion

L. glaucences Kuth leaves and its metabolite pinocembrin can be used to treat listeriosis due the bactericidal activity against L. monocytogenes.

Introduction

Foodborne diseases are caused by consuming food or beverages contaminated with viruses, bacteria, parasites, toxins, metals, and prions. Nowadays, foodborne diseases are an important public health problem around the world since; each outbreak has a series of direct and indirect costs, affecting public health, the economy and international food trade (Hoffmann & Scallan Walter, 2020). According to World Health Organization (WHO, 2023), in 2019, at least 1 in 10 people became sick and 33 million individuals with healthy lifestyles have died annually due to foodborne diseases which are particularly dangerous for children under <5 years old (1/3 young children’s deaths) or individuals with a weakened immune system. WHO also identifies, Listeria monocytogenes and Sallmonella as two of the main etiological agents of foodborne diseases.

Listeria monocytogenes is a foodborne opportunistic bacterial pathogen and is the causal agent of listeriosis, a disease that can be fatal to susceptible individuals (Jung, Yum & Jeong, 2017; Disson, Moura & Lecuit, 2021). Listeriosis is characterized by a wide spectrum of infections, which are categorized into two forms: severe invasive listeriosis and non-invasive febrile gastroenteritis. The first, is one of the most serious foodborne diseases, occurs in immunocompromised individuals and manifests itself as sepsis, meningitis, endocarditis, encephalitis, meningoencephalitis, septicemia and brain infection. While non-invasive gastroenteritis causes septicemia, atypical meningitis and febrile gastroenteritis accompanied by headache and backache (Matle, Mbatha & Madoroba, 2020).

Most cases of listeriosis are caused by consumed food products contaminated with L. monocytogenes (Duze, Marimani & Patel, 2021), which can survive and proliferate over a broad range of environmental conditions (low pH, high salt concentration, refrigeration temperature), as well as sublethal concentrations of biocides. This bacterium can be found in water, soil, food products, vegetables, meat, fish, seafood, ready-to-eat food, processed food, milk and dairy products. In addition, there are reports of L. monocytogens strains tolerant to biocides used in food processing and antibiotics, increasing the cases of listeriosis worldwide (Duze, Marimani & Patel, 2021), which pose a threat to food safety and public health (Baquero et al., 2020). For this reason, the search for new antilisterial drugs is necessary.

Litsea glaucescens Kuth is an aromatic tree endemic to México and Central America. Its common name is Mexican Bay-leaf and is known as “laurel” in Spanish. The leaves of L. glaucescens are commonly used in this area for food seasoning, replacing the leaves of the European species Laurus nobilis (Lauraceae), but they are also used in traditional medicine to treat diarrhea, vomit, bone pain, postpartum baths, colic in children, and illnesses related to the central nervous system (Guzmán-Gutiérrez et al., 2012). Due to its extensive use, L. glaucescens is one of the main non-timber forest products in México (Guzmán-Gutiérrez et al., 2012; Dávila-Figueroa et al., 2016). Several researchers have evaluated the antihypertensive, antidepressant, antioxidant, and antibacterial activities of this plant (Meckes et al., 1995; Guzmán-Gutiérrez et al., 2012; Cruz et al., 2014; Guzmán Gutiérrez, Reyes Chilpa & Bonilla Jaime, 2014; Gamboa-Gómez et al., 2016; Medina-Torres et al., 2016; López-Romero et al., 2018; Shi, Zhang & Guo, 2018; López-Romero et al., 2022). Regarding its antibacterial activities, studies have demonstrated the activity of this plant against some Gram-positive and Gram-negative bacteria, but its activity against L. monocytogenes has not been evaluated until now. The aim of this work was to produce different L. glaucescens extracts to test their effect against L. monocytogenes and to identify the compound responsible for the antibacterial activity.

Materials & Methods

Plant material

Litsea glaucescens leaves were collected in June 2019 from Cuautepec de Hinojosa, Hidalgo, México and were identified by Edith López Villafranco, head of the Herbarium at the Faculty of Higher Education Iztacala from Universidad Nacional Autónoma de México. A voucher sample was deposited in the herbarium with the code number 2533IZTA, then the leaves were dried under dark conditions at room temperature for three weeks. Afterwards, the plant material was ground using an electric blender.

Preparation of extracts

The dried, ground material (2.4 kg) was extracted consecutively by maceration with n-hexane (LgHex), dichlorometane (LgCH2Cl2), ethyl acetate (LgEtOAc), and methanol (LgMeOH) for 24 h, three times. All extractions were performed using 1:3 plant material/solvent ratio. The solvent was eliminated under reduced pressure distillation with a rotary evaporator (Büchi, Flawil, Switzerland).

Methanolic extract fractionation

The LgMeOH extract (18 g) was subjected to open column chromatography (60 x 680 mm) packed with silica gel 60 (mesh 70-230, 540 g) (Merck, Boston, MA, USA), and eluted with n-hexane/EtOAc/CH2Cl2/MeOH gradient system (100:0:0:0, 95:05:0:0, 90:10:0:0, 80:20:0:0, 60:40:0:0, 0:0:100:0, 0:0:70:30, 0:0:60:40, 0:0:50:50, 0:0:0:100). The volume of all samples was 500 mL. One hundred forty-two fractions were obtained, which were grouped into ten final fractions (C1F1 to C1F10) according to their chemical composition.

Flavanone detection and identification

Fraction C1F4 (1 g) was fractionated using open column chromatography (30 × 200 mm) previously packed with 30 g of silica gel 60 (mesh 70-230) (Merck) and eluted with a n-hexane/acetone (80:20, 79:21, 79:21, 78:22, 77:23, 74:26, 70:30) system. The volume of all samples was 10 mL. Fifty-five fractions were obtained, which were grouped into six final fractions (C2F1-C2F6) according to their chemical composition. Spectroscopic data for 1H, 13C, COSY, and HSQC NMR of C2F4 was performed in a Bruker Avance III HD 500 MHz NMR Spectrometer (Bruker, MA, USA). A mixture of CDCl3:CD3OD (1:1 v:v), as well as DMSO-d6 (Sigma-Aldrich, St. Louis, MO, USA) were used as a solvent.

Antibacterial assay

Bacterial strain

Listeria monocytogenes (ATCC19113) was donated by Javier Castro Rosas (ICBI, UAEH) and was used to test the antibacterial activity of the extracts, fractions, sub-fractions, and the purified compound.

The bacterial strain was cultivated in Muller-Hinton (MH) agar (Oxoid Ltd., Basingstoke, UK) at 37 °C. For the test, a bacterial inoculum was prepared under the National Committee for Clinical Laboratory Standards guidelines. Direct colony suspensions of overnight subcultures were diluted in MH broth (Difco, San Jose, CA, USA) and were adjusted to a 0.5 McFarland turbidity standard (approximately 108 colony-forming units [CFU]/mL).

Agar diffusion method

Antimicrobial activities of the extracts and fractions were evaluated by agar diffusion assays according to Mhalla et al. (2017) and Hoekou et al. (2017) with slight modifications. The plant extracts were prepared at 100, 50, 25 and 12.5 mg/mL, while the fractions were diluted to 10 mg/mL, and Kanamycin and Tetracycline (PanReactAppliChem, Darmstadt, Germany) at 0.032 mg/mL were used as positive control; finally, all dissolved samples were filtered in a sterile filter unit Millex®GV of 0.22 µm (Merck Millipore Ltd, Ireland). LgHex, LgAcOEt, and the fractions C1F3 to C1F9 were dissolved in acetone, LgCH2Cl2 and LgMeOH were dissolved in acetone:ethanol (1:1; v/v), C1F1 was dissolved in n-hexane, C1F2 was dissolved in n-hexane:acetone (1:1; v/v), and Kanamycin and Tetracycline were dissolved in sterile water. The sterile filter paper disks (six mm diameter) (Whatman, Maidstone, Kent, UK) were impregnated with the sample for the evaluation, then the solvent was allowed to evaporate from the extract-laden discs, and three extract/fraction/controls-treated discs were placed on a plate that was inoculated with fresh cell suspension (108 CFU). The negative control comprised the solvent used to dissolve the extracts and fractions. The plates were then incubated at 37 °C for 24 h. The diameters of the inhibition zones produced by the plant extracts, fractions, and controls (including the disk) were measured and recorded. All experiments were carried out in triplicate.

Minimal inhibitory concentration (MIC)

The active test compounds with inhibition zones and 5,7-dihidroxyflavanone were further investigated to determine their minimal inhibitory concentration (MIC) using a microdilution method (Morales-Ubaldo et al., 2022). Briefly, in 96-well plates, the stock solutions of the extracts were serially diluted twofold in methanol:water (1:9 v/v) to final concentrations between 200 to 0.195 mg/mL for extracts, 12.5 to 0.006 mg/mL for C1F1 to C1F10 fractions and 6.14 to 0.006 mg/mL for C2F1 to C2F6. Then, 100 µL of the inoculum (108 CFU/mL) were added to the wells. A sterility check (medium and solvent), negative control (medium, solvent, and inoculum), and positive control (medium, solvent, inoculum, and Kanamycin and Tetracycline) were included for each experiment. The plates were then incubated at 37 °C for 24 h at 70 rpm. After incubation, 20 µL of the INT salt (p-iodonitrotetrazolium chloride, 2 mg/mL) (Sigma-Aldrich) were added to each well and the plates were incubated at 37 °C for 30 min at 70 rpm. Bacterial viability was observed by the formation of pink color after the addition of INT. The MIC of each compound was established as the lowest concentration that completely inhibited the visible bacterial growth.

Minimum bactericidal concentration (MBC)

The minimum bactericidal concentration (MBC) was performed according to Morales-Ubaldo et al. (2022) with slight modifications. Briefly, 10 µL of each sample from the plates where there was no INT color change were added to the agar plates inoculated with a fresh cell suspension (108 CFU). Kanamycin and tetracycline at 32 µg per well were used as positive controls. The negative control comprised the vehicle where the sample was dissolved. These preparations were incubated at 37 °C for 48 h at 70 rpm. The MBC corresponds to the lowest concentration in which no growth was detected. All experiments were performed in triplicate. Moreover, the ratio of MBC/MIC of each sample was calculated to assess the antibacterial power.

Statistical analysis

The results obtained from the agar diffusion method were analyzed with an ANOVA, followed by a post-hoc Tukey test. Values of p < 0.01 were considered significantly different.

Results

The LgHex, LgAcOEt, LgCH2Cl2, and LgMeOH extracts were 24.0, 16.8, 52.8, and 76.8 g/kg of dry matter, respectively. The chromatographic fractionation of the LgMeOH extract allowed us to obtain nine fractions’ groups (C1F1 to C1F9). Further, C1F4 chromatographic purification afforded six final fractions (C2F1 to C2F6).

The TLC analysis of C2F4 indicated the presence of a pure compound, which was a white amorphous solid with mp = 111 °C. This compound was identified as pinocembrin (1) by NMR spectral data analysis. The proton nuclear magnetic resonance (1H NMR) spectrum of 1 on CDCl3-CD3OD (Table 1) showed signals for two aromatic units, one ABX aliphatic system, and two hydroxy groups. The multiple signal that integrates for five protons is ascribable to a monosubstituted benzene ring, while the AB system signals at δH 5.93 (1H, d, J = 2.1 Hz) and 5.94 (1H, d, J = 2.1 Hz) were characteristic of a 1,2,3,5 tetrasubstituted benzene moiety.

Table 1 1H and 13C NMR data of Pinocembrin (1) in different solvents.

	1 (This work)
CDCl3-CD3OD	1 (This work)
DMSO-d6	1 from Tanjung, Tjahjandarie & Sentosa (2013). (Acetone- d6)	1 from Napal, Carpinella & Palacios (2009). (DMSO-d6)	1 from Nyokat, et al. (2017). (CDCl3)	
	δ H	δ C	δ H	δ H	δ C	δ H	δ C	δ H	δ C	
2	5.36 (1H, dd, J = 13.0, 2.8 Hz)	79.26	5.39 (1H, dd, J = 12.5, 3.1 Hz)	5.49 (1H, dd, J = 4.0, 12.0 Hz)	–	5.44 (1H, dd, J = 3.2, 12.8 Hz	80.17	5.43 (2H, t, J = 6.7 Hz)	79.18	
3ax	3.01 (1H, dd, J = 17.1, 13.0 Hz)	43.39	3.05 (1H, dd, J = 17.1, 12.6 Hz)	3.06 (1H, dd, J = 12.0, 14.0 Hz)	–	3.06 (1H, dd, J = 12.8, 17.2 Hz)	40.45	3.10	43.37	
3eq	2.73 (1H, dd, J = 17.2, 3.0 Hz).	–	2.59 (1H, dd, J = 17.1, 3.1)	2.78 (1H, dd, J = 4.0, 14.0 Hz)	–	2.77 (1H, dd, J = 17.2, 3.2 Hz)	–	2.86	–	
4	–	195.89		–	197.3	–	196.75	–	195.65	
5	–	166.32		–	165.4	–	164.41	–	164.34	
6	5.93 (1H, d, J = 2.1 Hz)	96.68	5.68 (1H, d, J = 1.5 Hz)	5.86 (1H, d, J = 2.0 Hz)	–	5.52 (1H, d, J = 2.2 Hz)	96.84	6.03	96.81	
7	–	167.13		9.75 (1H, br, s, 7-OH)	168.5	–	167.62	–	–	
8	5.94 (1H, d, J = 2.1 Hz)	95.80	5.71 (1H, d, J = 1.5 Hz)	5.92 (1H, d, J = 2.0 Hz)	164.7	6.01 (1H, d, J = 2.2 Hz)	95.94	6.03	95.56	
9	–	164.06		–	–	–	163.59	–	163.11	
10	–	102.57		–	–	–	102.69	–	102.97	
1′	–	138.62		–	140.4	–	139.59	–	138.42	
2′	7.32–7.410 (5 H, m)	126.30	7.21–7.34 (5H, m)	7.44 (2H, m)	127.3	7.41 (5H, m)	127.47	7.41–7.49, m	126.16	
3′	7.32–7.410 (5 H, m)	128.96	7.21–7.34 (5H, m)	7.57 (3H, m)	129.7	7.41 (5H, m)	129.46	7.41–7.49, m	128.87	
4′	7.32–7.410 (5 H, m)	128.96	7.21–7.34 (5H, m)	7.57 (3H, m)	129.6	7.41 (5H, m)	129.39	7.41–7.49, m	128.87	
5′	7.32–7.410 (5 H, m)	128.96	7.21–7.34 (5H, m)	7.57 (3H, m)	129.7	7.41 (5H, m)	129.46	7.41–7.49, m	128.87	
6′	7.32–7.410 (5 H, m)	126.30	7.21–7.34 (5H, m)	7.44 (2H, m)	127.3	7.41 (5H, m)	127.47	7.41–7.49, m	126.16	
C-5-OH	11.99 (1H, brs)	–	11.95 (1H, s, D2O exchange)	12.20 (1H, brs, OH-C-5)	–	–	–	12.07	–	

The aliphatic signals at δH 2.73 (1H, dd, J = 17.2, 3.0 Hz), 3.01 (1H, dd, J = 17.1, 13.0 Hz), and 5.36 (1H, dd, J = 13.0, 2.8 Hz) were assigned to the ABX system formed by the protons H-3eq, H-3ax, and H-2, respectively of the cyclohexane ring of the flavanone, which were confirmed by COSY spectrum (Fig. S5). 1H NMR showed a signal for one hydroxy group [δH 11.99 (1H, brs, OH-C-5). The 1H NMR spectrum on DMSO-d6 showed signals consistent with those described by Napal, Carpinella & Palacios (2009) (Table 1). In addition, the presence of the hydroxyl at C-5 was confirmed by exchange with D2O of the signal at δH 11.95 (Table 1, Fig. S2).

13C NMR and DEPT spectra of 1 showed twelve separated signals, including one oxymethine (δC 79.26), one aliphatic methylene (δC 43.39), one ketone carbonyl (δC 195.89), four aromatic methines (δC 95.8, 96.68, 126.3, and 128.9), three oxyaryl carbons (δC, 164.06, 166.32, 167.13) and two aryl quaternary carbons (δC 102.57, 138.62). The HSQC spectrum (Fig. S6) showed correlations between the signals at δC 128.9 and 126.30 with a multiple signal at δH 7.32–7.41, indicating that these belong to the monosubtituted benzene ring. Integration of the NMR data obtained, indicated that this compound is a 5,7-dihydroxyflavanone. Comparison of the NMR data with those reported for pinocembrin (1) on different solvents (Napal, Carpinella & Palacios, 2009; Tanjung, Tjahjandarie & Sentosa, 2013; Nyokat et al., 2017), helped to its identification (Table 1, Fig. 1).

Figure 1 Key 1H- 1H-COSY and HSQC correlations of 5,7-dihydroxyflavanone (1).

The antibacterial test showed that the LgHex, LgCH2Cl2, and LgEtOAc extracts of L glaucescens were inactive within the tested concentration range, and only the methanolic extract of L. glaucescens (LgMeOH) exhibited antibacterial activity against L. monocytogenes (inhibition zone = 11.5 ± 0.45 mm, MIC = 4.5 mg/mL, MBC = 4.5 mg/mL) (Table 2). All fractions obtained from LgMeOH (CF1 to C1F9) were submitted to a pharmacological antibacterial test. As shown in Table 3, the C1F4 fraction was the only one with antibacterial activity (MIC = 0.78 mg/mL, MBC = 0.78 mg/mL). Further, C1F4 chromatographic purification afforded six final fractions (C2F1 to C2F6), and only C2F4 showed activity against the bacteria (MIC = 0.68 mg/mL, MBC = 0.68 mg/mL) (Table 4). In addition, the ratio MBC/MIC obtained for all samples was 1.

Table 2 Antibacterial activity of Litsea glaucescens leaf extracts against L. monocytogenes.

Extract	Concentration (mg/mL)	IZ (mm)
(Mean ± SD)	MIC (mg/mL)	MBC
( mg/mL)	R MBC/MIC	
LgHex	100	0	ND	ND	ND	
50	0	
25	0	
12.5	0	
LgCH2Cl2	100	0	ND	ND	ND	
50	0	
25	0	
12.5	0	
LgAcOEt	100	0	ND	ND	ND	
50	0	
25	0	
12.5	0	
LgMeOH	100	11.5 ± 0.45a	4.5	4.5	1	
50	0	
25	0	
12.5	0	
Kanamicyn	0.032	24.5 ± 0.05b	0.001	0.001	1	
Tetraciclyne	0.032	33.8 ± 0.1c	0.004	0.004	1	
Notes.

LgHex hexane leaf extract of L. glaucescens

LgCH2Cl2 dicholorometane leaf extract of L. glaucescens

LgAcOEt ethyl-acetate leaf extract of L. glaucescens

LgMeoH methanol leaf extract of L. glaucescens

IZ inhibition zone

MIC minimal inhibitory concentration

MBC minimum bactericidal concentration

ND not determinate

SD standard deviation

Different superscript indicate mean values that are significantly different (P < 0.01).

Table 3 Antibacterial activity of fractions obtained from Litsea glaucescens methanol extract (LgMeOH) against L. monocytogenes.

Fraction	Concentration (mg/mL)	IZ (mm)
(Mean ± SD)	MIC (mg/mL)	MBC
( mg/mL)	R MBC/MIC	
C1F1	10	0	ND	ND	ND	
C1F2	10	0	ND	ND	ND	
C1F3	10	0	ND	ND	ND	
C1F4	10	12.5 ± 0.04 a	0.78	0.78	1	
C1F5	10	0	ND	ND	ND	
C1F6	10	0	ND	ND	ND	
C1F7	10	0	ND	ND	ND	
C1F8	10	0	ND	ND	ND	
C1F9	10	0	ND	ND	ND	
C1F10	10	0	ND	ND	ND	
Kanamicyn	0.032	24.5 ± 0.05 b	0.001	0.001	1	
Tetraciclyne	0.032	33.8 ± 0.1 c	0.004	0.004	1	
Notes.

C1F1 to C1F10 fractions obtained from chromatographic purification of methanol leaf extract of L. glaucescens (LgMeOH).

IZ inhibition zone

MIC minimal inhibitory concentration

MBC minimum bactericidal concentration

ND not determinate

SD standard deviation

Different superscripts indicate mean values that are significantly different (P < 0.01).

Table 4 Antibacterial activity of sub-fractions obtained from C1F4 fraction of Litsea glaucescens against L. monocytogenes.

	C2F1	C2F2	C2F3	C2F4	C2F5	C2F6	Kanamicyn	Tetraciclyne	
MIC (mg/mL)	>6.14	>6.14	>6.14	0.68	>6.14	>6.14	0.001	0.004	
MBC
(mg/mL)	>6.14	>6.14	>6.14	0.68	>6.14	>6.14	0.001	0.004	
R MBC/MIC	ND	ND	ND	1	ND	ND	1	1	
Notes.

C2F1 to C2F6 fractions obtained from chromatographic purification of C1F4 fraction from methanol leaf extract of L. glaucescens (LgMeOH).

MIC minimal inhibitory concentration

MBC minimum bactericidal concentration

ND not determinate

Discussion

Despite the advances of the international community in food safety, foodborne diseases are still a serious public health problem. Furthermore, this problem is influenced by various factors, such as changes in eating habits, climate change, and resistance to antibiotics (Jung, Yum & Jeong, 2017). Listeria monocytogenes is recognized as one of the most important foodborne pathogens and is the causal agent of listeriosis, a disease that is caused by eating contaminated food that can be serious and is often fatal in susceptible individuals (Shi, Zhang & Guo, 2018). In humans, listeriosis treatments are hampered by the intracellular location of Listeria and the poor intracellular penetration of some antibiotics (Marini et al., 2018). Therefore, research into new antibacterial agents is required. Plant extracts, as well as the pure compounds obtained from them, are an important source of new antibacterial agents that are safe for the environment, humans, and animals (Kim et al., 2017).

The in vitro antibacterial activity of L. glaucescens extracts against L. monocytogenes was qualitatively evaluated by the presence or absence of inhibition zones toward the tested bacteria after MIC and MBC were determined. Only the most polar extract, LgMeOH, showed activity against L. monocytogenes (Table 2). This result agrees with other works where alcoholic extracts from this plant have shown antibacterial activity. Indeed, a methanolic extract of L. glaucescens leaves possess activity in vitro against Escherichia coli and Staphylococus aureus (CMI ≥ 1 and 0.8 mg/mL, respectively) (Meckes et al., 1995; López-Romero et al., 2018). Furthermore, Cruz et al. (2014) determined the MIC values of an ethanolic extract of L. glaucenses against Bacillus subtilis ATCC6051 (0.16 mg/mL), Mycobacterium smegmatis ATCC607 (0.62 mg/mL), and E. coli (0.62 mg/mL). These results suggested the presence of secondary metabolites in the alcoholic extracts of L. glaucescens leaves capable of damaging different bacterial strains. Thus, in this work, a LgMeOH bioguided chemical fractionation was performed to identify the active compounds that inhibit the growth of the most important foodborne pathogen Listeria monocytogenes.

LgMeOH purification produced ten final fractions, but only C1F4 showed activity against L. monocytogenes (inhibition zone = 12.5 ± 0.04 mm; CMI = 0.76 mg/mL). The TLC analysis of C1F4 showed a mixture of compounds, and for this reason, a subsequent purification was performed to obtain six new fractions (C2F1 to C2F6). All fractions were tested for their antibacterial capacity and only C2F4 showed antilisterial activity (Table 4). C2F4 was identified as 5,7-dihydroxyflavanone, known as pinocembrin, through a comparison of its NMR data (1H, 13C, COSY, HSQC) with data reported in the literature for this compound (Table 1). This flavonoid was previously isolated from the ethanolic extract of Litsea glaucescens bark by López et al. (1995), and it has been isolated from different sources such as Flourensia colepis (Napal, Carpinella & Palacios, 2009), Kaempferia pandurate (Tanjung, Tjahjandarie & Sentosa, 2013), Artocarpus odoratissimus (Nyokat et al., 2017) and Boesenbergia rotunda (Potipiranun et al., 2018).

Pinocembrin is one of the main flavonoids used in the pharmaceutical industry because of its antibacterial, antiparasitic, anti-inflammatory, antioxidant, antiapoptotic, anticancer, antifibrotic, hepatoprotective and neuroprotective biological activities. In addition, it can be used as a base skeleton for the synthesis of other flavonoids (Shen et al. 2019; Elbatreek et al., 2023). Regarding its antibacterial activity, there are no reports of the effect of this molecule against L. monocytogenes. However, pinocembrin has been effective against other Gram-positive bacteria (S. aureus, methicillin and gentamicin-resistant S. aureus, S aureus subsp. aureus Rosenbach, S. aureus penicillase (+), S. epidermidis, B. cereus, B. subtilis, and S. lentus and Streptococcus mutans), whose MIC values range from 0.001 to 0.5 mg/mL (Alcaráz et al., 2000; Drewes & van Vuuren, 2008; Katerere et al., 2012; Omosa et al., 2014; Joray et al., 2015; Veloz, Alvear & Salazar, 2019; Hernández Tasco et al., 2020). Besides, this compound caused the total disappearance of S. aureus populations at 1 mg/mL (Parra et al., 2016).

Despite L. monocytogenes being a Gram-positive bacteria, 5,7-dihydroxyflavanone possesses lower antibacterial activity against it (MIC = 0.68 mg/mL) compared to the effect described for other bacteria of the same type. Nevertheless, it is the first report of the activity of pinocembrin against L. monocytogenes.

Finally, to identify the type of effect exerted by L. glaucescens and pinocembrin, the MBC/MIC ratio was determined. An MBC/MIC ratio with values between 1 and 2 indicates bactericidal power, while ratios >2 indicate bacteriostatic activity (Btissam et al., 2018). Thus, according to the values obtained in this work for methanolic extracts and pinocembrin (ratio = 1) (Tables 2 and 3), L. glaucescens possess bactericidal power against L. monocytogenes and 5,7-dihydroxyflavanone is the compound responsible for it.

The antibacterial mechanism of different phenolic compounds has been thoroughly investigated. There are several forms in which flavonoids affect bacteria. They can inhibit the synthesis of nucleic acid and porins on the cell membrane, affecting energy metabolism, disturb cytoplasmic membrane function, reduce cell attachment and biofilm formation, change the membrane permeability, and attenuate of the pathogenicity (Farhadi et al., 2018). Furthermore, antibacterial agents can easily destroy the bacterial cell wall of Gram-positive bacteria, causing a leakage of the cytoplasm and its coagulation (Tian et al., 2018).

Soromou et al. (2013) reported that pinocembrin reduces α-hemolysin production, attenuated α-hemolysin mediated cell injury at low concentrations and protects mice from S. aureus-induced pneumonia; moreover, this flavanone increased cell permeability in Campylobacter jejuni, altering the metabolism (mainly protein production, redox cycle, and iron metabolism) (Klančnik et al., 2019; Elbatreek et al., 2023), and affected protein and DNA metabolism in Aeromonas hydrophila (Wu et al., 2022).

It is possible that the antibacterial effect of pinocembrin against L. monocytogenes is due to some of the mechanisms mentioned previously; however, subsequent studies are required to know which of them are responsible.

Conclusions

Although there are studies on the antibacterial activity of the alcoholic extracts of L. glaucences, the compounds responsible have not been identified. Some authors attributed this biological effect to the presence of flavonoids in the plant; however, this has not been clarified until now.

Our study could be considered the first to document the antilisterial activity of Litsea glaucences leaves and the isolation of the antibacterial agent in detail. In this work we evaluated the activity of four L. glaucences extracts against L. monocytogenes, the bioguided chromatographic separation of the methanolic extract allowed us to identify the 5,7-dihydroxyflavanone (pinocembrin) as the compound responsible for the antibacterial activity. In addition, the bactericidal effect against L. monocytogenes was demonstrated.

Supplemental Information

Supplemental Information 1 Inhibition zones

All fractions obtained from chromatographic separations. Each data is the diameter (mm) obtained for each treatment.

Click here for additional data file.

Supplemental Information 2 Supplemental Figures

The figures show all one and two dimension spectra obtained for Pinocembrin.

Click here for additional data file.

Supplemental Information 3 Chromatographic separation system

Click here for additional data file.

Additional Information and Declarations

Competing Interests

Author Contributions

Data Availability

The authors declare there are no competing interests.

Carlos David Gress-Antonio conceived and designed the experiments, performed the experiments, analyzed the data, prepared figures and/or tables, authored or reviewed drafts of the article, and approved the final draft.

Nallely Rivero-Perez conceived and designed the experiments, prepared figures and/or tables, and approved the final draft.

Silvia Marquina-Bahena performed the experiments, prepared figures and/or tables, and approved the final draft.

Laura Alvarez conceived and designed the experiments, prepared figures and/or tables, authored or reviewed drafts of the article, and approved the final draft.

Adrian Zaragoza-Bastida performed the experiments, authored or reviewed drafts of the article, and approved the final draft.

Víctor Manuel Martínez-Juárez conceived and designed the experiments, performed the experiments, authored or reviewed drafts of the article, and approved the final draft.

Carolina G. Sosa-Gutierrez performed the experiments, authored or reviewed drafts of the article, and approved the final draft.

Juan Ocampo-López analyzed the data, prepared figures and/or tables, and approved the final draft.

Armando Zepeda-Bastida analyzed the data, prepared figures and/or tables, and approved the final draft.

Deyanira Ojeda-Ramírez conceived and designed the experiments, performed the experiments, analyzed the data, authored or reviewed drafts of the article, and approved the final draft.

The following information was supplied regarding data availability:

The raw measurements are available in the Supplementary File.

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
