# Peer review of "Litsea glaucescens Kuth possesses bactericidal activity against Listeria monocytogenes"

_PeerJ, doi:10.7717/peerj.16522_

## Round 0.1 · original submission · Major Revisions

The article in its current form cannot be published. Queries made by both reviewers should be addressed and the journal format should be followed and the manuscript needs significant grammatical revisions. The paper could have been structured better with all the findings related to structure elucidation work which will certainly strengthen the findings. The co-relation of H-H ( COSY), and H-C ( HSQC) is to be presented with the molecule structure.

**Language Note:** The review process has identified that the English language must be improved. PeerJ can provide language editing services - please contact us at copyediting@peerj.com for pricing (be sure to provide your manuscript number and title). Alternatively, you should make your own arrangements to improve the language quality and provide details in your response letter. – PeerJ Staff

Reviewer 1 ·

Basic reporting

The research work intends to determine the effect of four extracts from leaves of L. glaucescens against L. monocytogenes and to perform bioassay-guided isolation of the active extracts and structural elucidation of the antibacterial compounds. The four extracts refer to four different solvents used ( DCM, n-Hexane, MeOH, Ethyl acetate) and this above sentence is to be corrected accordingly.

The language used in the whole article is smooth and simple to understand the objective.
Literature references used against the research work are relevant.
The tables and raw data supported the findings described in the research experiment model.
The results which were presented were relevant.

Experimental design

The primary objective of the research work reflects the activity against the bacteria and research results presented that LgMeOH fraction shows promising results against L. monocytogenes.
The methods and results described are presented with sufficient details.

The structure elucidation of 5,7-hydroxiflavanone described in the manuscript needs more experimental data.

Validity of the findings

All the results and literature cited are relevant to the findings.
Except for structure elucidation which needs more experimental data to be shared/included.

Structure elucidation of 5,7-hydroxiflavanone lacks the following details :
1. The 1H-NMR to be recorded in DMSO-d6 - to identify the presence of hydroxyl groups followed by D2O exchange to confirm these are exchangable protons.
2. LCMS and MSMS to be recorded to confirm the structure.
3. 2D-NMR data - correlations w.r.t adjacent protons to be presented - along with structure.

All the NMR data to be included in the manuscript - 1H-NMR 9 dmso-d6), 13C, COSY,, HSQC.

Without these data the structure elucidation work is incomplete.

Additional comments

The paper could have been structured better with all the findings related to structure elucidation work which will certainly strengthen the findings.
The co-relation of H-H ( COSY), and H-C ( HSQC) is to be presented with the molecule structure.

Reviewer 2 ·

Basic reporting

a) The manuscript needs significant grammatical revisions. For example, “foodborne illness (line 50)”; “In addition, them can be 56 deadly, especially…” (Line 55-56), etc.
b) References in the texts are non-uniform and do not follow general referencing format. Example, “Hoffmann and Scallan Walter, 2020” from line 52 & 53 may be written as “Hoffmann and Walter, 2020”, “Disson, Moura & Lecuit, 2021 “from line 61 & 62 may be written as “Disson et al., 2021”, etc.
c) It is a general rule that scientific names be italicized when typed written (Line 80). I may miss out some others.

Experimental design

a) Sequence of experimental steps from plant material collection to methanolic extract fractionation is found to be in correct order.
b) “Spectroscopic data for 1H, 13C, COSY, and HSQC NMR of C2F4 was their performance in a Bruker Avance III HD 500 MHz NMR Spectrometer (Bruker, MA, USA)” from line 122 – 124 has no logical meaning that I can comprehend and may otherwise be written as “was performed”.
c) “2.3” from line 127 may be deleted as it does not contribute to the meaning of the sentence being composed.
d) In Agar diffusion method (line 144), it was mentioned that paper disks of 6 mm diameter were used for extracts and fractions, but in line 148 for Kanamycin and Tetracycline of 32 µg/wells were used. This posed the question why disks were used only for impregnating extracts but not for kanamycin and tetracycline?
e) Results are not in order of materials and methods. Antimicrobial test results may come after isolation and structure elucidation results. Thus, rearrangement of the order of the results may be performed.
f) Photo of TLC of fraction C2F4 may be provided with solvent system, Rf value and method of derivatization used to validate the work.
g) Structural elucidation of pure compound, especially new compound, is not conclusive without High Resolution Mass Spectrometry, elemental analysis and infrared spectroscopy.
h) NMR spectral data (both H NMR and C NMR) of pure compound may be provided to validate the result given.
i) It was mentioned only in the discussion that the spectral data of the isolated compound was compared with prior reports from literature. This may be clearly mentioned in materials and methods and results. The word “isolation” in materials and methods may be changed to “detection” as I feel the results are inconclusive without the production of evidences in the form of High Resolution Mass Spectrometry, elemental analysis and infrared spectroscopy.

Validity of the findings

Antimicrobial activity results clearly support the aim of the study. However, isolation and structure elucidation may be looked into with a more critical perspective.

Additional comments

a) The manuscript requires major grammatical revision before being accepted for publication.
b) Referencing may be edited following the journal’s referencing style.
c) Scientific names should be italicized and when used again, the short form of the scientific name may be used.
d) If the authors could produce the TLC of fraction C2F4 with the solvent system used, Rf value and method of derivatization, as well as NMR spectral data (both H NMR and C NMR) of the pure compound deemed isolated, the manuscript is suitable for publication.

---

## Round 0.2 · accepted · Accept

The authors have addressed all of the reviewers' comments, this manuscript is now ready for publication.

Reviewer 1 ·

Basic reporting

My comments have been addressed adequatly.

Experimental design

My comments have been addressed adequatly.

Validity of the findings

No comments

Additional comments

No comments

Reviewer 2 ·

Basic reporting

Satisfactory

Experimental design

Satisfactory

Validity of the findings

Satisfactory

Additional comments

None